# Feasibility of Patient Reported Outcome Measures in Psychosocial Palliative Care: Observational Cohort Study of Hospice Day Care and Social Support Groups

**DOI:** 10.3390/ijerph192013258

**Published:** 2022-10-14

**Authors:** Natasha Bradley, Christopher Dowrick, Mari Lloyd-Williams

**Affiliations:** 1Centre for Health & Clinical Research, University of the West of England, Bristol BS16 1DD, UK; 2Department of Primary Care and Mental Health, University of Liverpool, Liverpool L69 3BX, UK

**Keywords:** palliative care, hospice day care, social support, patient reported outcome measures

## Abstract

Palliative care patients can be at risk of social isolation or loneliness. Interventions that can provide effective social support, and particularly emotional support, could facilitate healthy coping that bolsters quality of life and reduces depression in palliative care patients. This is an observational cohort study which recruited thirty patients (*n* = 30) from the day services of four independent hospices in England. Participants completed patient reported outcome measures in perceived social support, loneliness, and depression, at up to three time points. Age range was 56–91 years, males and females were equally represented, and the sample was 93% white British. In participants that provided two or more timepoints, perceived social support increased, and loneliness and depression decreased. Largest changes with the least variation between participants was in emotional support (*p* = 0.165) and loneliness (*p* = 0.104). These results suggest that the psychosocial patient reported outcome measures used (MOS-SS, UCLA, BEDS) could be sensitive to change aligned with the goals of this intervention in palliative care. Participants in this study were observed to derive psychosocial benefit from attending the hospice day service.

## 1. Introduction

Palliative care aims to prevent and relieve the suffering of people with life-limiting illness and their families, responding simultaneously to their physical, psychological, spiritual, social, cultural, and situational needs. Declining physical health and mobility can reduce opportunities to gain social support, leading to unmet social needs that might contribute to feelings of isolation, loneliness, or alienation [1] Guidance for patient care is to encourage the maintenance of existing social networks where possible, and to facilitate contact with other patients to lessen the pain of patient loneliness [2].

Hospices are prominent providers of palliative care in the United Kingdom, supporting more than 225,000 people each year [3]. Most hospices are independent charities working within and alongside the healthcare system, with varying remits and stated aims. Hospice day services facilitate social support for palliative care outpatients, in conjunction with other clinical or professional inputs, with the intention to improve quality of life so that beneficiaries can ‘live well’ [4]. Referral is often via a healthcare provider, such as by a General Practitioner or Hospital Consultant, but many hospices also accept self-referrals. Outcome measurement is not widely embedded into these social settings such that there is limited evidence available to identify best practice and inform decision-making.

A systematic review of group interventions in palliative care found that perceived social support had rarely been considered as an outcome or process variable [5]. It has been argued previously that social outcomes should be considered in the evaluation of hospice day care [6]. This paper reports on an observational cohort study using patient reported outcome measures in four hospice day services offering social support to patients. 

Social connection is an essential human need that is tied to our survival [7]. Meta-analyses have demonstrated social support and loneliness to be predictive of future morbidity and mortality in the general population [8,9,10]. Epidemiological reviews report loneliness to be both a consequence of chronic ill-health, and a risk factor for poor health in the future [11]. Chronic stress, loneliness, and depression have synergistic effects that promote inflammation which heightens future stress responses and worsens health [12]. Conversely, absent or decreasing loneliness predicts good self-rated health in the future [13].

Social isolation and loneliness are especially problematic in people with reduced mobility or physical function, which often confers decreasing economic and social resources [14]. Patients with respiratory disease and inadequate social support report more depression and worse health-related quality of life than those with high levels of social support [15]. Loneliness is associated with lower help-seeking intentions and higher rates of depression over time [16]. Patients with advanced cancer who report high social support have been reported to be less likely to develop depression [17]. Furthermore, patients with heart failure who report high social support, particularly emotional support, have greater confidence for self-care and symptom management [18].

Social wellbeing appears to be a foundation stone for good quality of life, but less is known about the outcomes of social support interventions, particularly within palliative care. Palliative day care is reported to improve patient mood [19], emotional wellbeing [20] and hope [21]. Hospice patients with depression have been reported to benefit from a narrative intervention that encourages them to ‘tell their story’ [22]. Providing a supportive environment for encouraging emotional expression and cognitive reflection in a narrative intervention can have a lasting impact (6+ weeks) on depression in palliative care [23].

Learning non-clinical everyday coping strategies from fellow patients reportedly improves psychological wellbeing, by helping to foster acceptance and encourage communication with others [24]. Sharing informational support in the form of ‘tips’ for managing symptoms, treatment, and the health system could help people to feel more in control of the illness experience [25]. Peer support could therefore encourage self-management as well as hope for the possibility of positive change [26,27,28]. Peer support in palliative care could be an effective intervention, but many studies are cross-sectional or descriptive, focussed on oncology, and reaching participants who are mostly well-educated, middle-aged, and female-therefore, questions remain on what forms of support are most effective and for whom [29]. 

Meeting other people with similar experiences, sharing emotional and informational support, and being free to act how you feel-these are social support processes that enable psychological adjustment to change. However, quantitative demonstration and meaningful definition of these processes is challenging in practice [30,31]. There is limited quantitative evidence on the effectiveness of hospice day services, and measurement of the social support expected to arise within these interventions has been almost absent [5,30].

Hospice day services are considered a complex intervention with multiple components, multi-disciplinary input, and a diversity in offering between hospices. It is not expected that there is consistency in the intervention(s) received, or the outcomes experienced by patients. Measurement of distinct but interacting concepts such as social support and loneliness is not straightforward, especially when concerned to minimise participant burden.

Sensitivity to change of outcome measures-their ability to detect change that may be occurring because of an intervention-is a reported challenge in this population. Focusing solely on health-related quality of life as a primary outcome measure may be inappropriate in palliative care because fluctuations in health can obscure the effect of intervention. As consensus is lacking on what constitutes patient benefit, outcomes measured are highly variable [32]. Progress in outcome measurement is needed, so that we are better able to identify ‘quality’ and compare service model effectiveness.

### Aim

This study tested the use of psychosocial patient reported outcome measures in hospice day services. The aim of this research was to establish an observational cohort study of psychosocial outcomes in palliative care patients attending a hospice day service. The research sought to establish whether change was observed over time. A secondary aim was to consider the feasibility and acceptability of using patient reported outcome measures in perceived social support, loneliness, and depression.

## 2. Materials and Methods

### 2.1. Study Design

An observational cohort study, conducted between August 2018 and May 2019. Data collection used patient-reported-outcome-measures at up to three timepoints during a hospice-based intervention offering social support.

### 2.2. Settings

Research locations were the day services at four independent hospices in England. Selection of research locations was informed by a survey of hospice day services undertaken by the research team [4]. A detailed understanding of each hospice day service was sought by the researcher (NB) during a concurrent qualitative component of research involving non-participant observations and service-provider interviews. Some hospices intended to begin recruitment but were unable to recruit new patients (reasons cited: current service redesign, personnel changes, low referrals of new patients).

Table 1 provides brief information on the four hospice day services involved in recruiting for this study. Site 1 offered traditional hospice day care, with facilitated circle discussions on illness-related and one-to-one clinical input provided in side-rooms. Site 2 was similar in many respects, but had a greater emphasis on spirituality and nature, and more resources for art, including both patient groups and family art therapy. Site 3 provided a mixture of closed and open group settings, so that a patient might attend the time-limited exercise group in the morning and choose to stay for lunch, then join an art group in the afternoon. Site 4 also used open access activities and tailored support groups, using an appointment model only for clinical input.

‘Traditional’ here means full day attendance on a specified day, usually with transport and care provided. Both these sites offered time-limited attendance, but a step-down programme was in formal operation at site 2. ‘Non-traditional’ day services are characterised here by flexibility-patients have a greater degree of choice between different open access components and in when they access the service. Both sites 3 and 4 aimed to limit use of patient transport, but a small number of patients continued to receive organised transport and the additional care required for their level of clinical need. Through the arrangement of transport these patients had a specified day to attend the service.

### 2.3. Participants 

Participants were recruited when they were about to start or had recently started attending day services. Participants were adults with life-limiting illness, currently living in the community (i.e., not inpatients or care home residents). Patients were eligible to participate if they had adequate English language skills to participate in the research methods and they did not have impaired cognition that would limit their ability to give informed consent (e.g., due to dementia). 

### 2.4. Recruitment 

Hospice service-providers identified eligible patients and invited them to participate in the research after they had been assessed, invited, and agreed to attend the hospice day service. This meant that patients met the diagnostic and functional criteria for attendance at the hospice day service, before being assessed for inclusion in this study.

Potential participants were provided with study information and the opportunity to ask questions either of the hospice staff or the researcher, and to take the information home to discuss with others, before giving written consent. NB visited each research location to begin patient recruitment with service-providers.

### 2.5. Sample Size 

A three-month recruitment window was initially agreed with each hospice, with a review at the end of this period to decide whether to continue to recruit for a further 3 months. Due to sparse available literature in this area [5], ethical approval was initially obtained for 100 participants for a feasibility study.

### 2.6. Data Collection

Stakeholder input during research design or feasibility work can usefully inform the selection of sensitive patient reported outcome measures for this context [33]. In this study, the selection of outcome measures was discussed early on with hospice staff and a ‘declined to answer’ box was added to all questions. This strategy was intended to indicate the acceptability of the measures used and to reduce the extent of missing item-level data. The questions from the four outcome measures (discussed below) were printed together as a question pack. The patients were asked if they were able to complete this independently and they were provided help only if required, by the researcher (NB) or a hospice staff member or volunteer.

Hospice staff provided brief reasons for attrition on behalf of participants where possible, because information on the nature of missing data and cause behind attrition (e.g., death or withdrawal) is useful to inform most appropriate imputation methods for larger data sets [34]. Data collection began on the first week of a person’s attendance at a hospice day service and generally repeated at 4–6 weeks and 10–12 weeks. Emphasis on research design was on flexibility to plan for the fluctuating health and therefore attendance of many day hospice patients, by avoiding strict dates for data collection with the intention to reduce missing data [34,35]. This meant there was variability between participants in the length of time that participants were observed. A data collection end date was agreed with all research locations.

Patient reported outcome measures used were: Medical Outcomes Study Social Support Scale (MOS-SS) for perceived social support [36]; University of California & Los Angeles (UCLA) 3-item scale for loneliness [37]; Brief Edinburgh Depression Scale (BEDS) for depression in advanced illness [38]; and EuroQol’s EQ5D-5L for health-related quality of life [39] (Table 2). As a proxy for health resource use, two questions on A&E attendance and overnight hospitalisations were included. In total, there were 36 questions in this question pack. Brief demographic and clinical information (e.g., diagnoses) were provided by the patient at the first timepoint.

#### 2.6.1. Perceived Social Support

Social support measures can be overly long and might not distinguish between functional and structural social support, or include support received from outside of the family, couple, or healthcare encounter [40]. The Medical Outcomes Study was a large two-year study of chronically ill patients, for which a relatively brief multidimensional self-administered social support scale was developed (MOS-SS) [36]. Questions ask for perceived availability of different types of social support-emotional, informational, tangible, affectionate, and positive social interaction-to construct an overall social support index. This five-factor model of functional social support has been shown to have validity in cancer patients [41]. It is distinct from measures of structural social integration and from measures of negative loneliness feelings [42].

#### 2.6.2. Loneliness

The concept of a lonely person carries negative connotations that might make it difficult for individuals to recognise and admit to feelings of loneliness in themselves. A suggested consequence of this is that measuring loneliness with direct questions such as ‘do you feel lonely?’ leads to underreporting, whereas loneliness measures that have multiple ‘indirect’ questions can have better reliability [43].

The UCLA loneliness scale [44] is commonly used and has been revised and adapted for several contexts and countries. Its earlier format included twenty questions with a mixture of direct and indirect questions, reflecting positive and negative social experiences. Participant burden was a concern if using this scale, which is relatively lengthy. A simplified and shorter version contains three indirect questions related to feelings of lacking companionship, being left out, or feeling isolated, with validity demonstrated in population-based surveys [37] and in breast cancer survivors [45]. The brief UCLA scale focuses on loneliness as a subjective experience, without using the word ‘lonely’. There are three questions, each with three possible answers, giving a minimum possible score of 3 and maximum score of 9.

#### 2.6.3. Depression

Several depression measures, such as the Hospital Anxiety and Depression Scale (HADS) and PHQ-9, ask questions about symptoms that could have both a physical and emotional cause (e.g., tiredness, dizziness) and can therefore be difficult to complete by patients in poor health. Somatic symptoms are highly influenced by disease load in advanced illness patients [46]. Using only non-somatic depressive symptoms can differentiate advanced cancer patients with depression from those without [47]. The Brief Edinburgh Depression Scale (BEDS) was designed for use in palliative and supportive care, with the recognition that physical symptoms of depression and chronic illness are difficult to untangle-thus it focuses on cognitive and emotional symptoms only [48].

#### 2.6.4. Quality of Life

The EQ-5D-5L represents a Likert-style scale from no problems to extreme problems in each area of mobility, self-care, usual activities, pain/discomfort, and anxiety/depression. Each dimension in this measure represents one question scored 1 to 5 by the participant. 

The five domains of the EuroQoL’s EQ-5D-3L are considered relevant and recognisable to patients, but the -3L measure may have poor responsiveness to clinical change, and not just in palliative care contexts [49]. Consequently, a five-level version has been developed and tested with five possible answers for each domain, rather than three [39]. Comparing EQ-5D-3L with the newer 5L version indicates that the scale is more informative and shows fewer ceiling effects [50]. Improved discriminatory power and responsiveness to clinical improvement have been reported in COPD and diabetes [51,52]. EQ-5D-5L has also been used in cost-consequence analysis of hospice day services [53].

### 2.7. Analysis

Analysis focused on comparing groups of participants by differences at baseline and exploring change over time for participants with two or more timepoints. Independent samples Kruskal–Wallis test was used to compare differences between groups at baseline by reason for attrition and research location. To consider change over time, analysis tested within subject differences for each outcome measure using Wilcoxon signed rank test in SPSS. A univariate model was used to consider if change over time varied by whether participants lived alone or with others, and by gender.

The intention of this study was to observe and compare patient outcomes over three timepoints, however overall sample size was small and especially so at T3. Recruitment in most cases relied on referrals into the hospice from statutory healthcare services, which were received at an inconsistent rate. Furthermore, the uncertainty of the disease and its progression makes it challenging to predict accrual and attrition rates to a research study. Reporting is focused on T1 and T2 due to the small sample at T3. 

## 3. Results

### 3.1. Participants

Thirty (*n* = 30) patients were recruited across four research locations (Table 3). Data collection is depicted in Appendix A. Recruitment is difficult in palliative populations and the sample size was determined by the success of recruitment across the sites in the recruitment period. 

At T1 (*n* = 30), the sample was 50% male, 93% white British, and age range was 56-91. By T2 (*n* = 19), age range was consistent, but sample was 53% male and 100% white British. At T3 (*n* = 5), sample was 60% female, and age range was 58–81. Information on prognosis was not collected. Seven participants died during data collection (23.3%) and seven stopped attending the day service due to their declining health (23.3%).

The majority of this sample (73%, *n* = 22) reported a cancer diagnosis, most commonly lung, breast, prostate, and bone. Fourteen participants (47%) reported a non-cancer diagnosis-nine participants (*n* = 9) had both a cancer diagnosis and a non-cancer diagnosis; five participants (*n* = 5) had a non-cancer diagnosis without cancer. Common non-cancer diagnoses in this sample were chronic obstructive pulmonary disease (COPD), heart failure, Parkinson’s disease, and arthritis. 

### 3.2. Recruitment

Recruitment at each site occurred on a rolling basis, with the first site starting recruitment in August 2018 and all sites finishing recruitment by the end of April 2019. Data collection for all sites ended in May 2019. The intention of using a recruitment window rather than a fixed number of participants per site was to reduce issues of selection bias, as it was intended that all eligible patients starting to attend the service within the recruitment window would be invited to take part. It was difficult to elucidate how closely this procedure was followed. Notably, a defined time window for recruitment appeared more acceptable than a target number for the service-providers involved.

### 3.3. Data Collection

Most participants completed question booklets for themselves, within the hospice. In sites 1 and 2, timing of data collection fitted in with patient assessments which meant that the procedures of the setting supported follow up. Two participants required assistance to read the questions, comprehension appeared straightforward. In site 4, some participants took their question booklets home to complete but did not return them. When timepoints were completed, almost all participants completed every question, suggesting the questions themselves were not too burdensome. However, two participants declined to complete the visual analogue scale of the EQ-5D-5L.

### 3.4. Reasons for Attrition

Reporting on the type and cause of missing data assists the interpretation of palliative care research [35]. In this study, death or declining health was the cause of most non-completion. The frequency of reasons for attrition is shown in Appendix A. Participants in this sample were less likely to complete the study if they were from more deprived postcodes (Appendix A). Older patients were more likely to have an unknown attrition reason (Appendix A). Psychosocial measures differed by reasons for attrition: particularly, participants who died during data collection reported higher perceived social support (*p* = 0.768), lower depression (*p* = 0.761), and less loneliness (*p* = 0.169) at baseline than other groups (Figure 1).

### 3.5. Quality of Life (EQ-5D-5L)

Table 4 summarises the frequency of problems in each dimension as reported by participants, which is useful to describe health-related quality of life in this sample. Correlations between timepoints were significant for all domains (*p* < 0.1). Problems reported in the domain of self-care increased between the first and second timepoint (*p* = 0.012). Trends were for problems in other domains to decrease, but these were not statistically significant. Differences between research locations were not significant. There were differences in self-care by reasons for attrition between those who left the study by choice and those who left the study due to poor health (*p* = 0.105).

### 3.6. Perceived Social Support (MOS-SS)

The mean change for overall social support was positive, but this was not a significant difference (Table 5). Change was not observed in the affectionate support subscale: the mean change is (slightly) negative; and the standard deviation is lowest for this domain, indicating the least variation in change between participants. The emotional social support subscale showed the largest mean change between the two timepoints, a high t-value, and *p*-value < 0.5. This demonstrates that change between first and second timepoint was largest in the domain of emotional support. Change over time in perceived social support and in the emotional support subscale is depicted in Figure 2. 

A similar trend was seen in the small sample at T3 (*n* = 5). Mean change between first and third timepoint was greater for emotional support (11.5) than for overall social support (8.8) (Appendix A). 95% confidence intervals are wide, reflecting this is not robust, but within subject differences over time were approaching significance in a univariate model of emotional support (*p* = 0.165) (and not over social support (*p* = 0.381)).

### 3.7. Loneliness (UCLA)

Loneliness was observed to decline between T1 and T2 (*n* = 19) and this was approaching statistical significance (*p* = 0.104) (Figure 3). Seven participants reported decreasing loneliness, nine participants maintained their loneliness scores, and three participants reported increasing loneliness. The change in loneliness and depression scores between T1 and T2 (*n* = 19) is shown in Table 6 (see also Appendix A).

### 3.8. Depression (BEDS)

Depression was observed to decline between T1 and T2, but variation was high, and the finding was not statistically significant (Figure 4, Table 6). Eight patients reported decreasing depression, seven reported increasing depression, and four remained at the same score.

At baseline, eighteen participants (60%) had a score of 6 or above, indicating clinically significant depression. Proportions were similar at the second timepoint, when thirteen participants (68.4%) reported a score of 6 or above. At the third timepoint (*n* = 5), remaining, there were two participants reporting score of 6 or higher. Four of the five participants who completed three time points indicated depression of clinical severity at baseline.

### 3.9. Research Location

Differences between research locations were not significant at baseline for any outcome measure (Appendix A). Differences in depression were significant at T2 (*p* = 0.025), with the most difference between sites 3 and 4, but this could be attributable to unequal attrition (Appendix A).

### 3.10. Patient Context

Participants that lived alone and completed two timepoints (*n* = 6) appeared to have a greater reduction in depression and loneliness, and a greater increase in perceived social support than those who lived with others (*n* = 13). Samples are small and low confidence is indicated by large error bars as shown in Appendix A.

Comparing change over time between men and women (*n* = 19) shows that men reported greater change in depression than in other measures, whereas women reported change in other domains but less in depression. These trends (Appendix A) suggest there could be gender differences in the support derived from an intervention or how it is observed using these measures. 

## 4. Discussion

Interventions that provide effective social support, and particularly emotional support, could facilitate healthy coping behaviours that bolster quality of life and reduce loneliness or depression in palliative care patients [17,22,54]. There is some evidence that mental health and physical health benefits of intervention in advanced illness are mediated by improvements in social domains [55,56] However, providing evidence for any change occurring as a result of these interventions remains challenging, partially due to ambiguity around intended goals and appropriate outcome measures.

This study recruited thirty patients from four hospice day services and change over time was observed between the first and second timepoint for nineteen patients (*n* = 19). Perceived social support increased, and loneliness and depression decreased. Most significant changes were in loneliness (*p* = 0.104) and in the emotional support sub-scale (*p* = 0.165). There were no indicators that the psychosocial outcome measure questions caused distress or were burdensome to complete.

Although attrition was high, item level missing data was low. The ‘declined to answer’ box was ticked by two participants, only in relation to the Visual Analogue Scale of the EQ-5D-5L These participants did not have apparent vision difficulties and answered the other questions in the outcome measures question pack. This suggests that the psychosocial patient reported outcome measures used (MOS-SS, UCLA, BEDS) could be sensitive to change aligned with the goals of these interventions and acceptable to at least some patients.

Findings align with other reports on how these social interventions can be beneficial to at least some patients in palliative care. Participant distress was significantly reduced during a palliative rehabilitation program that included group physiotherapy twice per week for 8 weeks [57,58]. In that sample, emotional concerns were as prevalent in the sample as physical concerns at baseline; and attrition was nearly a third over 8 weeks. It is suggested that social support within the group enables emotional expression that relieves the stigma of ‘losing the cancer battle’ [58]. A different palliative rehabilitation program offered group support for 12-weeks, in addition to individual appointments and self-help materials [59]. 45% of the invited participants chose the group format, and these participants reported more emotional problems at baseline than those preferring individualised content. Group interventions appear more acceptable to some participants than others, and this could be related to the extent of emotional difficulties experienced [57,59]

Personal context-including age, gender, ethnicity-shape the availability and need for different types of support, as well as, unfortunately, opportunities to access hospice services. Neighbourhood demographics and economic resources are associated with symptom burden, especially depression and pain, and unmet informational needs [60]. Research into the effectiveness of interventions may benefit from considering the differences in outcomes or experiences for patients from different contexts, particularly: people living alone, people with clinical depression at baseline, people of different ages or genders, and people from areas of deprivation.

There is a disconnect between what is known about the importance of naturally existing relationships and the knowledge available to guide and evaluate interventions that provide social support or alleviate loneliness [14,61]. Intervening to provide social support has the potential for detriment, particularly if social experiences inadvertently increase loneliness or decrease feelings of autonomy or competence. Loneliness could be aggravated by situations where the person perceives that they have little control over their interactions or feels they are instrumental to another person’s goal [8]. Detrimental social responses can confer a future reluctance to disclose emotional experiences, resulting in lower wellbeing despite the presence of social support [62]. We should be wary of rushing to provide ‘just any’ social ties-social conflict, even from well-intended actors, can result in rumination, poorer adherence to health behaviours, and other unhelpful coping strategies [63]. It is how social support is perceived, rather than the number of interactions, that influences coping behaviours and treatment adherence-quality over quantity [64].

Results here are presented with caution as the interpretation of effects from non-randomised studies requires larger samples. Recruitment challenges and high attrition is common in similar studies (e.g., attrition of one third over 8 weeks [57]). A sufficiently powered observational cohort study using these outcome measures could contribute to further theoretical development. Obtaining that large sample size over at least three timepoints would require planning for anticipated attrition with substantial time and resource commitment. 

This study is focusing specifically on individuals who are able to access hospice day services (meeting eligibility criteria and geographical catchment) and chose to do so (indicating some motivation to attend). It is possible that people accessing such services differ from the broader palliative population and these differences may influence their experience of the intervention. Qualitatively, it would be interesting to explore further how the ‘dose’ received within social interventions may be (at least partially) be moderated by participants ability and willingness to engage.

The research design was acceptable to the hospices involved and did not disrupt patients in the hospice support that they received-to them, it represented additional questionnaires, rather than a change to care trajectory. Early in-person meetings were helpful to understand potential concerns of managers or staff, and perhaps the team felt more engaged with the project because they had a ‘face to the name’. However, not all the hospices involved were successful in recruiting patients (due to: service redesign, retirements, low patient numbers). Personnel changes added to the unpredictability of recruitment-hospice day services may rely on a small number of staff roles which confers a large impact from one or two being off work for any reason. These outcome measures therefore appear to be acceptable, but feasibility may vary by organisational context.

## 5. Conclusions

The use of appropriate psychosocial outcomes can facilitate progress in demonstrating and comparing the effectiveness of interventions in palliative care, especially when an over-reliance on health-related quality of life or physical function would not give the full picture. This paper reports on the use of patient reported outcome measures to observe change over time for patients attending hospice day services.

Findings suggest social support in palliative care might improve emotional support and decrease loneliness for some patients. We have briefly discussed the challenges and opportunities of the method used. It appears that patient reported outcome measures of perceived social support, loneliness, and depression could be useful in social interventions in a palliative care or hospice setting.

## Figures and Tables

**Figure 1 ijerph-19-13258-f001:**
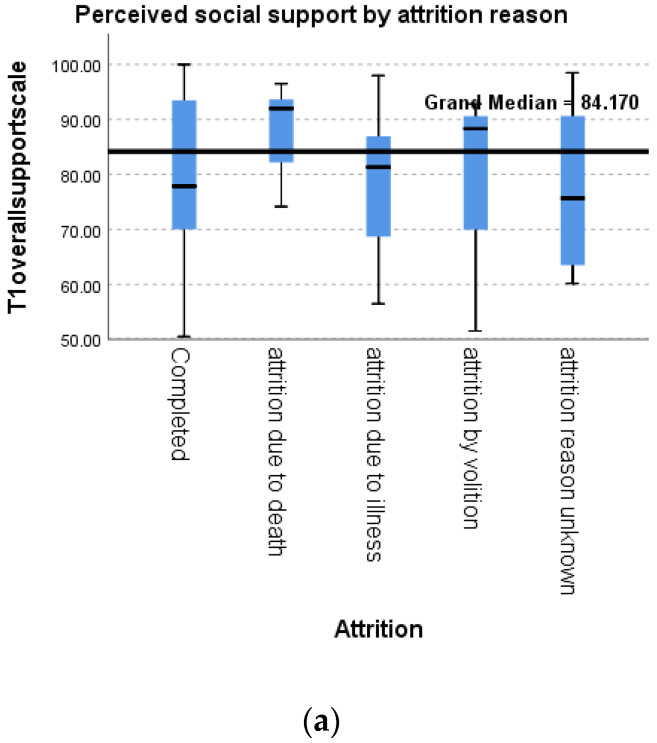
Psychosocial measures at baseline by reason for attrition: (**a**) Perceived social support; (**b**) Depression; (**c**) Loneliness. *: In a box plot, the interquartile range is shown as a box, and the median shown as a line. Asterisks are data points >1.5 times the median value.

**Figure 2 ijerph-19-13258-f002:**
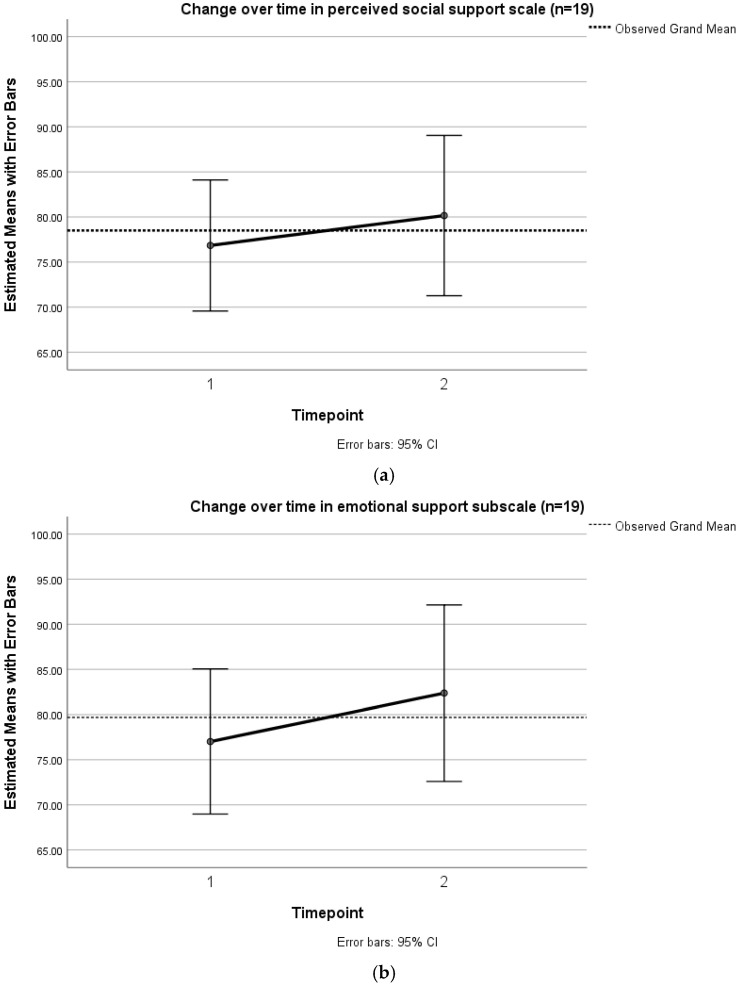
Change over time between T1 and T2 (*n* = 19): (**a**) Perceived social support; (**b**) Emotional support subscale.

**Figure 3 ijerph-19-13258-f003:**
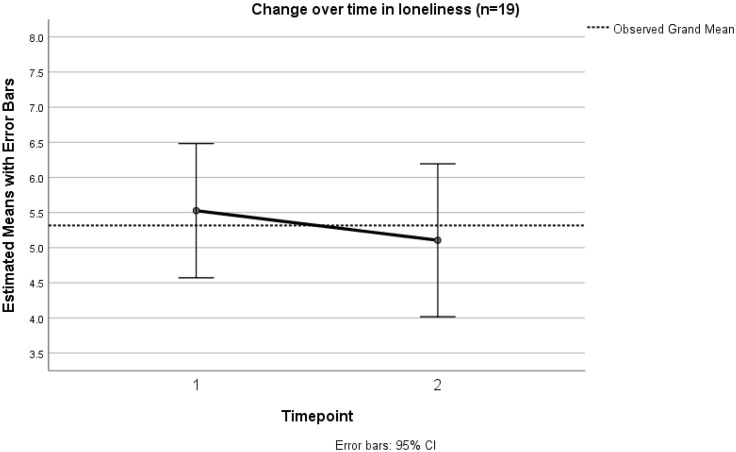
Change over time between T1 and T2 (*n* = 19) in loneliness.

**Figure 4 ijerph-19-13258-f004:**
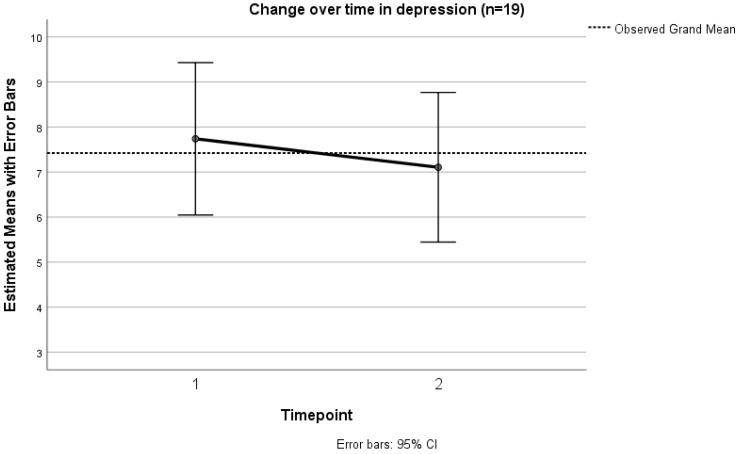
Change over time between T1 and T2 (*n* = 19) in depression.

**Table 1 ijerph-19-13258-t001:** Research locations. Note: index of multiple deprivation is according to hospice postcode (postal location).

Site	Recruitment Dates	Region of England	Index of Multiple Deprivation (0–10, 10 = Least Deprived)	Urban or Rural Hospice Location	Hospice Services (Intervention Description)
1	December 2018–March 2019	North-West	5	Urban city and town	Traditional day care with clinical input & facilitated discussions. Specified day to attend. Transport provided.
2	August 2018–March 2019	South-West	9	Rural village	Traditional day care with spirituality, art, and nature. Transport provided. Stepdown programme in operation.
3	February 2019–April 2019	South-East	5	Urban major conurbation	Rehabilitative exercise group, plus social & therapeutic programme. Open access social space. Limited transport provision.
4	October 2018–April 2019	South Coast	5	Urban city and town	Open access art, exercise, & wellbeing groups. Clinical input & transport by appointment.

**Table 2 ijerph-19-13258-t002:** Patient reported outcome measures used and rationale.

Domain.	Measure	Description	Reason for Selection
Perceived social support	MOS-SS	19 questions, inc. emotional, informational, tangible, affectionate, and interactional support	Questions about perception of different types of support, including outside of the family unit, and so reflects goals of intervention.
Loneliness	UCLA 3-item	3 indirect questions on loneliness, distinct from functional support and depressive symptomology	Brief measure of perception of relevant negative social experiences reflecting loneliness, without using word ‘lonely’.
Depression	BEDS	6 questions, based on cognitive and affective depressive symptoms	Brief measure suitable for advanced illness and validated for use in palliative care.
Quality of life	EQ-5D-5L	5 questions-mobility, self-care, usual activities, pain, anxiety/depression-plus visual analogue scale	5 domains commonly used in practice so familiar to participants, 5 L could be more sensitive to change than 3 L version.

**Table 3 ijerph-19-13258-t003:** Participant demographics and clinical features. Note: Diagnoses missing for one participant in site 3.

	Site 1	Site 2	Site 3	Site 4	Total
**N**	8	6	10	6	30
**Age**					
Range	58–77	56–87	56–91	59–84	56–91
Mean (standard error)	69.38 (2.764)	70.5 (5.542)	68.5 (3.769)	78.17 (3.894)	71.07 (1.999)
95% confidence interval	62.84–75.91	56.25–84.75	59.97–77.03	68.16–88.18	66.98–75.15
**Gender**					
Male	5	0	7	3	15
Female	3	6	3	3	15
**Ethnicity**					
White British	8	5	9	6	28
Asian Ugandan	0	1	0	0	1
Black Caribbean	0	0	1	0	1
**Diagnosis**					
Cancer	2	4	7	3	16
Noncancer	2	0	1	3	6
Multimorbidity inc cancer	4	2	1	0	7
**Living alone**	1	1	4	2	8

**Table 4 ijerph-19-13258-t004:** EQ-5D-5L results for the sample with significance tested for pairs (*n* = 19). Note: two participants declined to complete Visual Analogue Scale (VAS) of overall health ‘today’.

Dimension.	T1 (*n* = 30)	T2 (*n* = 19)	Significance
Mobility			
1	13.3%	5.3%	0.248
2	10%	15.8%	
3	53.3%	42.1%	
4	20%	36.8%	
5	3.3%	0%	
Self-care			
1	40%	26.3%	0.012
2	23.3%	21.1%	
3	20%	42.1%	
4	13.3%	5.3%	
5	3.3%	5.3%	
Usual activities			
1	13.3%	5.3%	0.273
2	10%	15.8%	
3	36.7%	31.6%	
4	26.7%	21.1%	
5	13.3%	26.3%	
Pain/discomfort			
1	13.3%	10.5%	0.248
2	6.7%	26.3%	
3	63.3%	47.4%	
4	16.7%	15.8%	
5	0%	0%	
Anxiety/Depression			
1	43.3%	42.1%	0.357
2	30%	26.3%	
3	23.3%	31.6%	
4	3.3%	0%	
5	0%	0%	
Overall health (VAS 0-100)	(*n* = 28)	(*n* = 19)	
Median	56	60	0.420
Mean	54.18	55.63	
Standard deviation	16.678	20.421	

**Table 5 ijerph-19-13258-t005:** Subscales of perceived social support (MOS-SS), change between T1 and T2 (*n* = 19).

	Mean Change	Standard Deviation	Standard Error of Mean	95% Confidence Interval of the Difference		T Value	*p* Value (2 Tailed)	Wilcoxon Signed Rank Test
Overall social support	3.316	16.082	3.689	−0.435	11.067	0.899	0.381	0.267
Emotional	5.357	16.114	3.697	−0.410	13.124	1.449	0.165	0.114
Tangible	5.263	24.970	5.729	−0.772	17.299	0.919	0.370	0.189
Affectionate	−0.351	13.050	2.994	−0.641	5.940	−0.117	0.908	0.904
Interaction	4.211	18.485	4.241	−0.699	13.120	0.993	0.334	0.268

**Table 6 ijerph-19-13258-t006:** Mean Change Between T1 and T2 in Loneliness and Depression.

	Mean Change	Standard Deviation	Standard Error of Mean	95% Confidence Interval of The Difference		T Value	*p* Value (2 Tailed)	Wilcoxon Signed Rank Test
Loneliness	−0.421	1.071	0.246	−0.937	0.95	−0.714	0.104	0.104
Depression	−0.632	2.985	0.685	−0.070	0.807	−0.922	0.369	0.528

## Data Availability

Data available on request from the corresponding author. The data are not made available publicly due to privacy concerns as participants could be identifiable within the small sample obtained.

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
