# Peer review of "Feasibility of Patient Reported Outcome Measures in Psychosocial Palliative Care: Observational Cohort Study of Hospice Day Care and Social Support Groups"

_ijerph, 2022, doi:10.3390/ijerph192013258_

Round 1
Reviewer 1 Report
I thank you for your work!
I only allow myself to provide some small advice:
1) You speak of 3 timepoints, and in the paper, you refer that the third was attended by just 5 persons, but all the Figures are concerned with only T1 and T2...probably you could explicitly declare how you intend to manage T3 in this research, to better explicate the results in lines 311... and 368...
2) Probably you can explicitly define the exact number of questions in line 150: 33 + 2 + Health scale;
3) Probably in line 186 you can add: (BEDS) just after having explained the acronym;
4) In Table 3: Column 'Site 3', you have to adjust the lines (the last 1 is not in the correct line) and the sum (it is 10 and not 9)...
5) In Table 4: Column T1 'Overall health', why there is (n=28) and not 30?
The last suggestion concerns the possibility, in this study or in a further one, to relate the different types (proposals) of social support (see Table 2 third column) with each betterment of patients.
Finally, as you notice, it could be very interesting to repeat the study with a larger number of patients probably both with general (as in this study) and with specific (as proposed) social supports, and repeated over more than three timepoints...but I understand the difficulties ...
Thank you again!
Author Response
Thank you for your comments and suggestions, they have been helpful in improving the paper.
- You speak of 3 timepoints, and in the paper, you refer that the third was attended by just 5 persons, but all the Figures are concerned with only T1 and T2...probably you could explicitly declare how you intend to manage T3 in this research, to better explicate the results in lines 311... and 368...
We have made small adjustments to ‘2.7. Analysis’, particularly lines 210-215.
We have inserted additional material into the supplementary files to demonstrate findings in the sample at T3. These are: -
- Probably you can explicitly define the exact number of questions in line 150: 33 + 2 + Health scale;
We have added that detail as a helpful indication of the burden of these outcome measures.
- Probably in line 186 you can add: (BEDS) just after having explained the acronym;
BEDS has been added to this section.
In Table 3: Column 'Site 3', you have to adjust the lines (the last 1 is not in the correct line) and the sum (it is 10 and not 9)...
Thank you – I have amended this formatting and added a note to clarify diagnosis is missing from one participant in site 3.
- In Table 4: Column T1 'Overall health', why there is (n=28) and not 30?
This is because two participants declined to complete the Visual Analogue Scale (EQ-5D-5L VAS). This is reported in the ‘3.3 Data collection’ but we have now added it as a note to the table.
The last suggestion concerns the possibility, in this study or in a further one, to relate the different types (proposals) of social support (see Table 2 third column) with each betterment of patients.
I agree that it is worth exploring how the different ‘types’ of social support included in this measure of perceived social support (emotional, informational, tangible, social interaction/companionship) might be experienced by the patient and have different outcomes. In this study, emotional support showed greater increase. This study, using patient-reported-outcome-measures, had a related qualitative component, which is in preparation for publication and looks in more detail at the processes or mechanisms by which social support might impact on patient outcomes.
Finally, as you notice, it could be very interesting to repeat the study with a larger number of patients probably both with general (as in this study) and with specific (as proposed) social supports, and repeated over more than three timepoints...but I understand the difficulties ...
Thank you. I certainly agree. I hope to attempt more work along these lines in the future.
Reviewer 2 Report
This is a report from an effort to explore the feasibility of creating a cohort of individuals participating in day palliative care. This care delivery model is not common in the U.S. It would help this reader if the authors could include a brief description of the pathway leading to this clinical option.
Since this is a feasibility study and the goal is cohort development, the literature review should focus on prior studies and their experiences / strategies with developing this type of cohort. Perhaps the delivery sites could be engaged and provide context data for the size of their participant pool.
If the goal of this work is evaluating the performance of the individual surveys in this context, then less interpretation of results (i.e. responses) and more details concerning the process of interviews would support this goal.
Ultimately, the manuscript is not clear about why they are doing this and what the expected results will be. This attached file contains comments boxes with specific questions.

Author Response
Thank you for providing such detailed and thoughtful feedback. We have responded below. We are appreciative of your feedback in the manuscript and have provided edits particularly to ‘Table 1’ and the methods section, to clarify the setting and procedure of this research. The information of how these particular outcome measures have been previously used is briefly reviewed in the materials and methods section.
Please note that palliative care is a broad discipline and not necessarily limited to the last days or weeks of life. Most hospices in the UK are independent charities who might provide support to patients with life-limiting illness and their families/caregivers for many years throughout their ‘illness journey’. People with terminal illness may have their lifespan extended through medical treatment but still have significant challenges associated with living with illness. They might cope with these challenges through healthy coping behaviours (such as honest communication, reappraisal, new and adapted activity, hope and gratitude; in contrast to unhelpful coping behaviours such as rumination, imposed isolation, alcohol and drug abuse). Healthy coping behaviours are those that help a patient ‘live well’ with illness. Psychosocial palliative care is specified as distinct from clinical input or interventions for symptom control – psychosocial palliative care prevents suffering by responding to psychological and social needs.
This is a report from an effort to explore the feasibility of creating a cohort of individuals participating in day palliative care. This care delivery model is not common in the U.S. It would help this reader if the authors could include a brief description of the pathway leading to this clinical option.
Thank you. We have extended the definition of palliative care and explanation of hospices in the introduction and added a helpful reference that gives extensive detail on hospice day services in relation to social support. We have also added much more information on the particular research settings and diagnoses included in this study.
Since this is a feasibility study and the goal is cohort development, the literature review should focus on prior studies and their experiences / strategies with developing this type of cohort. Perhaps the delivery sites could be engaged and provide context data for the size of their participant pool.
There are relatively few papers attempting patient reported outcome measure within hospice day services. We reference several systematic reviews and specific examples within the literature review. We outline how there are suggestions that this type of intervention is beneficial, but the existing evidence is weak. In palliative care, sensitivity of outcome measures and appropriateness of the outcomes measured in line with the goals of the intervention are identified as challenges.
To our knowledge, perceived social support has not been used previously as a patient reported outcome measure of an intervention in a hospice setting. The research design used here recruited patients as they were about to start attending the hospice group – therefore, we depended on a flow of new patients into the hospice, which was ultimately outside of the control of this investigation, or the research location.
If the goal of this work is evaluating the performance of the individual surveys in this context, then less interpretation of results (i.e. responses) and more details concerning the process of interviews would support this goal. Ultimately, the manuscript is not clear about why they are doing this and what the expected results will be. This attached file contains comments boxes with specific questions.
The aim of this research is established at the end of the introduction: to observe change over time using outcome measures aligned to the goals of intervention, and to establish whether these measures were acceptable and feasible. We were not able to obtain sufficient sample size to compare between research locations, but the experience of trialling these research methods (including adding a ‘decline to answer’ box) was valuable. These were patient-reported outcome measures that in general were completed by the patients themselves within the hospice using hard-copy question packs. We detail the type and cause of missing data caused by attrition and also by participants missing or declining any question. It did not appear that the questions were burdensome for patients to complete because we were able to account for most ‘drop-outs’. We also reflect on the research design experience within the discussion.
Reviewer 3 Report
Thank you for the opportunity to review this manuscript. Overall, it is well written, with only a sentence or two requiring further clarity for readability.
The study is interesting but as noted by the authors very small in terms of sample size, with attrition also problematic and therefore unsure how valuable this will be to those working in the sector in supporting palliative patients. It is unlikely to be of value in terms of evidence-based practice for example as a means of changing interventions.
Further detail about specific hospice services at each collection site would be useful to determine which of these is most effective in alleviating depression/loneliness etc. As would further clarity around diseases. While hospice primarily focuses on those with cancer, it was interesting to see that some with other conditions were included - what these were would certainly be of interest and which activities helped these individuals regarding the outcomes identified.
The rationale for flexible recruitment was positive part of this study, but given a year for recruitment across 4 institutions, numbers remain small. Why were no other hospices included in the study, or perhaps representation from the North East of England, given the other 'corners' of the country were included. Also, what was the rationale for including primarily mid to low decile areas? As loneliness and depression/social isolation can still occur in the affluent, well-educated individual. It would be useful to clarify this also.
Addressing these would improve the understanding of the sample and why the focus you have taken.
Author Response
Thank you for providing your insightful comments and helpful suggestions. We have responded below and within the manuscript as noted.
Thank you for the opportunity to review this manuscript. Overall, it is well written, with only a sentence or two requiring further clarity for readability. The study is interesting but as noted by the authors very small in terms of sample size, with attrition also problematic and therefore unsure how valuable this will be to those working in the sector in supporting palliative patients. It is unlikely to be of value in terms of evidence-based practice for example as a means of changing interventions.
Although the study with such small sample is not likely to inform evidence-based practice in and of itself, it is necessary work to inform future research that can have such value. Our previous systematic review found very few studies with palliative care patients (only one, in 2018) that had attempted to measure social support as an outcome of intervention, even when that intervention explicitly intended to increase social support. The extent of palliative care practice in the hospice sector that provided or facilitated social support is not yet well-evidenced, but this paper is a step in the right direction.
Further detail about specific hospice services at each collection site would be useful to determine which of these is most effective in alleviating depression/loneliness etc. As would further clarity around diseases. While hospice primarily focuses on those with cancer, it was interesting to see that some with other conditions were included - what these were would certainly be of interest and which activities helped these individuals regarding the outcomes identified.
Due to the small sample size and other limitations of the research design we do not argue in this paper that one research location is superior to another. A brief description of the hospice day service is provided in table 1, but we have added much more detail to describe the hospice services involved. We have also provided more information on the diagnoses included in this sample. We politely disagree that hospice focuses on cancer – this is a widespread belief and may remain the case in some places, but in practice there is increasing recognition of the palliative care needs associated with non-malignant conditions and initiatives aiming to reconcile disparity of access.
The rationale for flexible recruitment was positive part of this study, but given a year for recruitment across 4 institutions, numbers remain small. Why were no other hospices included in the study, or perhaps representation from the North East of England, given the other 'corners' of the country were included. Also, what was the rationale for including primarily mid to low decile areas? As loneliness and depression/social isolation can still occur in the affluent, well-educated individual. It would be useful to clarify this also. Addressing these would improve the understanding of the sample and why the focus you have taken.
Thank you. The length of recruitment period for each site varied between 3 and 9 months, which is detailed in table 1. It was intended that recruitment numbers would be higher and more research locations will be included – more detail is given below, for your interest. For the manuscript, we have added a comment that not all research locations initially involved in the project did recruit patients.
This stage of data collection includes hospices from different regions of the country, in small villages and large cities, serving different populations, with varied deprivation profiles. Research sites were identified using insights from the survey as described in the previous chapter. The chief executive officer, day patient services leaders, or head of research governance (where applicable) was approached with an invite to participate in a funded PhD project. Exact process differed in each site – in some cases a research committee formally reviewed and recommended the project, in others the researcher was able to meet with relevant decision-makers and explain the purpose of the research directly.
Hospice day services are complex interventions with multiple interacting components, and consistency in the intervention received or outcomes experienced are not expected between patients. An understanding of each service was achieved by the researcher during a concurrent qualitative investigation, which is described in the next chapter. Not all hospices invited were able to take part in recruiting patients, with reasons cited related to ongoing service redesign or changes in personnel, meaning lack of time to commit or uncertainty in patient availability. Many of these sites did take part in the qualitative investigation.
Recruitment windows are staggered because the researcher visited sites to conduct qualitative data collection and organise the start of recruitment. All sites had a minimum of one full day participant observation of services, up to 3 days when a large variety of services were available. Interviews were conducted with between 2 and 4 service providers in each recruiting site. Additional interviews and visits were also carried out in three other hospices, who were not able to recruit patients at the time due to current service redesign or the design of relevant services: two urban locations in the North-West (deciles 4 and 8) and a rural South-West site (decile 10).
Two other sites intended to begin recruitment in August 2018 and participated in interviews and observations – these were a rural site in the South-East (deprivation decile 6), and a rural site in the North-West (decile 7). These hospices were unsuccessful in recruiting after three months and withdrew from taking part in this section of the study; this was due to personnel changes in one site (maternity leave and retirement within a small staff team) and low patient numbers in the other (with referrals during the study period being sparse, when present they tended to be General Practitioners referring people in the last few weeks of life).
Round 2
Reviewer 2 Report
The authors have provided a detailed response to the comments in my review.